# Longitudinal Changes in CD4^+^, CD8^+^ T Cell Phenotype and Activation Marker Expression Following Antiretroviral Therapy Initiation among Patients with Cryptococcal Meningitis

**DOI:** 10.3390/jof5030063

**Published:** 2019-07-17

**Authors:** Alice Bayiyana, Samuel Okurut, Rose Nabatanzi, Godfrey Zziwa, David R. Boulware, Fredrick Lutwama, David Meya

**Affiliations:** 1Department of Immunology and Molecular Biology, School of Biomedical Sciences, College of Health Sciences, Makerere University, Kampala 7072, Uganda; 2Infectious Diseases Institute, College of Health Sciences Makerere University, Kampala 22418, Uganda; 3Makerere University Walter Reed Project, Kampala 16524, Uganda; 4Division of Infectious Diseases and International Medicine, Department of Medicine, University of Minnesota, Minneapolis, MN 55455, USA; 5Department of Internal Medicine, School of Medicine, College of Health Sciences, Makerere University, Kampala 7072, Uganda

**Keywords:** HIV, T cell phenotypes, ART, cryptococcal meningitis

## Abstract

Despite improvement in the prognosis of HIV/AIDS (human immunodeficiency virus/acquired immune deficiency syndrome) patients on antiretroviral therapy (ART), cryptococcal meningitis (CM) still causes 10–15% mortality among HIV-infected patients. The immunological impact of ART on the CD4^+^ and CD8^+^ T cell repertoire during cryptococcal co-infection is unclear. We determined longitudinal phenotypic changes in T cell subsets among patients with CM after they initiated ART. We hypothesized that ART alters the clonotypic phenotype and structural composition of CD4^+^ and CD8^+^ T cells during CM co-infection. For this substudy, peripheral blood mononuclear cells (PBMC) were isolated at four time points from CM patients following ART initiation during the parent study (ClinicalTrials.gov number, NCT01075152). Phenotypic characterization of CD4^+^ and CD8^+^ T cells was done using T cell surface marker monoclonal antibodies by flow cytometry. There was variation in the expression of immunophenotypic markers defining central memory (CD27^+^CD45R0^+^), effector memory (CD45R0^+^CD27^–^), immune activation (CD38^+^ and Human Leucocyte Antigen DR (HLA-DR^+^), and exhaustion (Programmed cell death protein one (PD-1) in the CD4^+^ T cell subset. In comparison to the CD4^+^ T cell population, the CD8^+^ central memory subset declined gradually with minimal increase in the effector memory subset. Both CD4^+^ and CD8^+^ T cell immune exhaustion and activation markers remained elevated over 12 weeks. The relative surge and decline in the expression of T cell surface markers outlines a variation in the differentiation of CD4^+^ T cells during ART treatment during CM co-infection.

## 1. Introduction

The high burden of human immunodeficiency virus (HIV)-related opportunistic infections remains an enormous challenge to optimal HIV care [1], with cryptococcal meningitis (CM) contributing 10–15% of HIV-related mortality in sub-Saharan Africa [2,3]. Cryptococcal meningitis is the leading opportunistic infection after tuberculosis despite considerable decline in HIV prevalence during the era of antiretroviral therapy (ART) [4]. The etiological agent, *Cryptococcus* spp., evades the host innate and adaptive T helper responses, which are critical for immune control of the opportunistic infection [5,6]. HIV infection leads to depletion and distortion of the CD4^+^ T cell repertoire causing lymphocyte defects, which renders the immune-suppressed host susceptible to disseminated cryptococcal infection [7,8,9].

Distortion of the CD4^+^ T cell repertoire affects the functionality and structural composition of the lymphocytes. Phenotypic markers such as CD45R0, CD27, CD28, CCR7, and cytokine production define T cell subsets [10], which are altered during HIV infection. Depletion of CD4^+^ T cells is associated with increased CD8^+^ T cell memory cells and activation, which are predictive of HIV disease progression [7]. However, immune restoration to optimal T cell counts by ART has been documented [11,12,13,14,15,16]. After initiation of potent therapy, the distribution of memory T cell subsets improves with subsequent antigen reduction [17]. A study among ART treated CM patients revealed greater *Cryptococcus neoformans*-specific immune responses were associated with improved survival and lower cerebrospinal fluid (CSF) fungal burden compared to the ART-naïve cohort [18]. Despite the efficacy of ART, T cell phenotypic abnormalities often persist, suggesting incomplete recovery. Some markers of immune activation and inflammation remain elevated while the magnitude of HIV-specific T cell immune responses contract [18]. Thus, absolute T cell counts alone may not reflect complete immune reconstitution. More recent work has shown a positive correlation between increased expression of markers on T cells, such as PD-1, and the level of viremia [19,20]. 

A study by Rallón et al. showed that incomplete CD4^+^ T cell restoration may be associated with defective restoration of the central memory T cells, a subset with a pivotal role in T cell homeostasis [21]. Additionally, an impaired immune response not only predisposes to co-infections but also influences the prognostic outcome [22]. Amino acid diversity within the T cell repertiore (TCR) repertoire is a qualitative feature of T cell responses [17,23]. Furthermore, different CD4^+^ and CD8^+^ T cell subsets are associated with specific outcomes in both infectious diseases and immune disorders [17,24], which implies that the T cell phenotype, in addition to the number of circulating CD4^+^ T cells, is crucial in determining the outcome of CM. There is a paucity of data on the effect of ART on T cell surface markers in HIV-associated CM [7,11,18].

We hypothesized that initiation of ART alters the structural composition and phenotype of CD4^+^ and CD8^+^ T cells among patients with CM. In this substudy, we evaluated the CD4^+^ and CD8^+^ T cell phenotypic changes among CM patients after they had initiated ART.

## 2. Materials and Methods

### 2.1. Study Design

This was a prospective study leveraging peripheral blood mononuclear cells (PBMC) samples collected from the Cryptococcal Optimal ART Timing (COAT) trial, (ClinicalTrials.gov number, NCT01075152) [14] that focused on optimal timing of ART initiation in HIV-positive patients with CM. Cryopreserved PBMC samples collected from participants at four different time points, i.e., at 0, 4, 8, and 12 weeks following CM diagnosis were used for this substudy.

### 2.2. Study Site Setting and Participants

Study participants were enrolled from Mulago National Referral Hospital, Kampala, Uganda from November 2010 through June 2012. Patients with suspected meningitis who provided individual or surrogate consent were screened. ART-naïve adults with an index episode of CM were subsequently enrolled. Cerebrospinal fluid (CSF) and PBMC were stored at baseline, 4, 8, and 12 weeks after enrolment. Participants were randomly assigned to initiate ART 7–11 days after CM diagnosis (early ART group) or 4–6 weeks after CM diagnosis (deferred ART group) following treatment guidelines [25].

### 2.3. Sample Size Estimation

The number of samples was determined using the WHO sample size calculator [26] at a 0.05 level of significance. The proportions in groups 1 (p1) and 2 (p2) were 0.25 and 0.75, respectively. The difference between proportions under the alternative hypothesis (H1) was 0.5. The sample size calculated accounting for attrition of 0.1 was 29.

### 2.4. PBMC Thawing and Stimulation

Cryopreserved PBMC samples were retrieved from liquid nitrogen at −196 °C and immediately transferred to a preset 37 °C water bath. Upon thawing, cells were washed in complete RPMI 1640 medium and then rested overnight at 37 °C in 5% carbon dioxide (CO_2_). After resting, we determined cell yield where viability testing was done using trypan blue solution. Cells were stained using 0.4% trypan blue solution at a 1:1 dilution ratio. Samples with at least 80% viability were used for stimulation assays. At least 2.5 × 10^6^ PBMC were used for stimulation to test antigen.

We prepared co-stimulatory antibodies; CD28a and CD49d from a stock of 1.0 mg/mL each by diluting in a sterile 50 Mm tris PH 8.6 solution to make a working stock of 10 g/mL. A 96-well plate, clearly mapped out was used. Using a pipette, 10 µg of CD28 (4 µL) in 1 mL of PBS and 10 µg of CD49d (2 µL) in 2 mL of PBS was added to each well and mixed thoroughly. Then, 200 µL PBMC aliquots were aseptically added into each well. The test antigen, cryptococcal glucuronoxylomannan (GXM), was also diluted 1:20 to achieve a working volume of 3.3 µL. For each sample, three wells were desginated, i.e., a nil control, test antigen, and 0.5 µL of positive control staphyloccal enterotoxin B (SEB). Plates were incubated for a total of 6 h at 37 °C in a CO_2_ incubator at a 5° slant. After the first 2 h of incubation, 20 µL of Golgistop was added and incubation was continued for more 4 h. After incubation, cells were washed with 200 µL staining buffer per well and then transferred to the 5 mm round bottomed polystyrene FACS tubes.

### 2.5. Cell Surface Staining

Subsequently, the cells were surface stained and incubated for 45 minutes with the following monoclonal antibodies; CD3^Amcyan^, CD4^Pacific blue^, CD8^APC H-7^, CD45R0^PerCP Cy5.5^, CD38^PE Cy7^ and PD-1^PE^, CD27^APC^, HLA-DR^FITC^ (BD Biosciences, San Jose, CA, USA). The cells were acquired on an eight-color FACS CANTO II (BD Biosciences, San Jose, CA, USA). At least 100,000 events were recorded for analysis. Gating was standardized and set using fluorescence minus one controls (FMOs). Data obtained were analyzed using FlowJo version 10.1 (San Carlos, CA, USA) and GraphPad Prism 7.0 (GraphPad Software Inc., La Jolla, CA, USA).

Table 1 above shows the cell surface markers used to investigate phenotype changes among the CD4^+^ and CD8^+^ T cell populations. We used these markers to obtain their respective monoclonal antibodies for staining and multiparameter flow cytometry.

### 2.6. Data Analysis

Flow cytometry data were analyzed using FlowJo version 10.1 software. Statistical analysis was performed using GraphPad Prism version 7. The frequencies of CD4^+^ T cells expressing activation and memory markers were measured as a percentage of the CD3 T cell parent cell population. Differences in the frequencies of the T cell subsets over time were evaluated using repeated measures analyses and Kruskal–Wallis test for non-parametric comparisons. The level of significance was set at a *p* value of ≤0.05.

### 2.7. Ethical Approval

The parent study was approved by the School of Medicine Ethics Review Committee at Makerere University (REF 2009-022, 29 January 2009). We obtained a waiver of consent from the Institutional Review Board of the School of Biomedical Sciences, Makerere University to use the stored data collected from the approved parent study.

## 3. Results

### 3.1. Participant Characteristics

In this study, we included 20 participants who were initiated on ART after the diagnosis of CM. Of these, 13 were male (65%), and the average age was 37 (±SD) years. Baseline characteristics of the study participants are shown in Table 2.

### 3.2. Gating Strategy

We developed a gating strategy and used it to analyze the cryptococcal-specific phenotypic and activation marker expression on CD4^+^ and CD8^+^ T cells. Figure 1 shows representative flow cytometry dot plots to identify the specific CD4^+^ and CD8^+^ T cell subsets from an HIV-1-infected participant with cryptococcal meningitis.

### 3.3. CD4^+^ and CD8^+^ T Cell Counts Increase Four Weeks after ART Initiation 

HIV-1 infection is characterized by depletion of CD4^+^ T cells [27] and an inversion of the CD4/CD8 T cell ratio. However, initiation of ART results in the reconstitution of CD4^+^ T cell counts [28]. Our initial aim in this study was to determine CD4^+^ and CD8^+^ T cell recovery after the start of ART over time. We did not observe a significant difference in CD4^+^ and CD8^+^ T cell counts four weeks after initiation on ART. However, there was a significant increase in CD4^+^ T cell counts from four weeks onwards. We conclude that HIV-1 and cryptococcal co-infected patients start to demonstrate adequate immune reconstitution four weeks post ART initiation.

In Figure 2, we noted a significant increase (*p* value of 0.01) in the CD4^+^ T cell count, four weeks after the start of ART. The frequency of the T cells is expressed as a percentage of the CD3^+^ T cell counts. On the scatter plots, the horizontal lines represent the median frequencies of the specific cells. The results shown are corrected for unstimulated responses. The Kruskal–Wallis test was used to compare differences in frequencies of CD4^+^ and CD8^+^ T cells at the four different time points. * represents a *p* value of 0.01. *p* values < 0.05 were considered significant. SCR represents Screening. A timepoint at which all participants were diagnosed positive for CM and started ART.

### 3.4. CD4^+^ T Cell Phenotypic Expression Profile

We hypothesized that ART alters the structural composition and clonotypic phenotype of the CD4^+^ T cell repertoire in HIV-associated CM. Results revealed a variation in the expression of the phenotypic markers giving rise to different subpopulations, i.e., effector and central memory T cell subsets. Contrary to the kinetics of CD4^+^ T cell subsets, the CD8^+^ T cell central memory subset declined gradually although this was not statistically significant (*p* = 0.227). The frequency of CD4^+^ T cells expressing CD45R0^+^CD27^+^ remained constant.

In regard to immune activation, there was a decline in expression of CD38^+^HLA-DR^+^ between week 4 and week 8 in both the CD4^+^ T and CD8^+^ T cells. However, this decline was not statistically significant. Changes in the frequency of CD4^+^ T cells expressing PD-1 reflected a slight decline, yet the CD8^+^ T cell frequency remained constant. The central memory subset was preserved with slight increases, which were not statistically significant, yet the effector memory subset declined.

Figure 3 shows slight declines in the CD4^+^ T cell central memory (T_CM_)subset, although not statistically significant differing from the CD8^+^ T_CM_ subset. T_CM_ subsets were shown by concurrent expression of CD27^+^ and CD45R0^+^. The horizontal lines on the scatter plot represent the median frequencies of the specific cells. The results are corrected for unstimulated responses. The Kruskal–Wallis test was used to compare differences in frequencies of CD4^+^ and CD8^+^ T cells at the four different time points. Dots correspond to individual determinations.

In Figure 4, we noted a significant decrease in the CD4^+^ T_EM_ from week 4 to week 12 unlike the CD8^+^ subset. Recovery of T_EM_ was shown by the expression of CD45R0^+^. The horizontal lines on the scatter plot represent the median frequencies of the specific cells. The results shown are corrected for unstimulated responses. The Kruskal–Wallis test was used to compare differences in frequencies of CD4^+^ and CD8^+^ T cells at the four different time points. * represents a *p* value of 0.02. *p* values less than 0.05 were considered significant. Dots correspond to individual determinations.

Figure 5 shows non-significant slight declines in levels of immune activation 12 weeks after ART treatment and CM diagnosis. Immune activation was measured as the frequency of dual expression of CD38^+^HLA-DR^+^ on CD4^+^ and CD8^+^ T cells shown in Figure 5A,B, respectively. The frequency of expression of CD38^+^HLA-DR^+^ was measured as a percentage count of the cells expressing the markers. The horizontal lines on the scatter plot represent the median frequencies of the specific cells. The results shown are corrected for unstimulated responses. The Kruskal–Wallis test was used to compare differences in frequencies of CD4^+^ and CD8^+^ T cells at the four different time points. Dots correspond to individual determinations.

In Figure 6, we noted slight declines in immune exhaustion, although not statistically significant. Expression of PD-1 on the CD4^+^ and CD8^+^ T cells represents immune exhaustion. PD-1 expression is shown as a percentage on CD4^+^ and CD8^+^ T cells in Figure 6. On the scatter plot, the horizontal lines represent the median frequencies of the specific cells. Results shown above are corrected for unstimulated responses. The Kruskal–Wallis test was used to compare differences in frequencies of CD4^+^ and CD8^+^ T cells.

## 4. Discussion

In this study of the kinetics of T cell populations among HIV patients co-infected with cryptococcal meningitis, we noted that over time the T cell counts increased, with declines in effector memory subset. Analysis of T cell recovery was combined with measurements of memory, activation, and exhaustion markers to establish to what extent ART had affected these parameters.

In a normal immune response, there is low T cell immune activation of the T and normal CD4^+^ T cell counts ranging from 500 to 1500 cells per mm^3^. HIV infection is characterized by the depletion of the CD4^+^ T cells, high CD8^+^ count and high immune activation. In presence of a CM co-infection, there is CD4^+^ T cell depletion, high immune activation, and decreased cryptococcal memory cells.

Initial descriptions delineate effector T lymphocytes as cells that increase in settings of active antigenic stimulation, and, upon secondary challenge and persistent antigenemia, T_CM_ lymphocytes differentiate into effector memory T cells [29,30].

The dynamics of CD4^+^ and CD8^+^ T cell turnover are influenced by different mechanisms that are accentuated during HIV infection [31,32,33,34]. Results showed that there was a general increase in the T cell counts after ART initiation throughout the 12 weeks of follow up. A similar observation was made by Butler et al. that the number of CD4^+^ T cells increases during the first few weeks of ART [33,35]. An increase in the T cell numbers could primarily be associated with re-distribution of the cells from the memory pool [36,37]. A study on T cell homeostasis suggested that early increases in CD4^+^ T cells comprise primarily of recirculating memory T cell populations, while later increases appear to be preferentially naïve T cells [38]. We could also postulate that the increase was partially due to proliferation as observed by a surge in HLA-DR expression over time. However, this would require using more proliferation markers. Proliferation of CD4^+^ T cells is controlled more tightly by CD4^+^ T cell numbers contrary to the case in CD8^+^ T cells.

On the contrary, there was reduction in the cell counts at week four. A possible explanation for this could be the presence of a cryptococcal co-infection to suggest that T cell recovery occurs after four weeks in HIV-associated CM [39]. Additionally, we postulate that individual host immune factors including baseline counts, thymic size, and genetic factors could influence the response to treatment [40,41]. Studies have established that up to 20% of patients may have suboptimal CD4^+^ T cell recovery despite HIV virologic suppression [21,36,42].

The relative number of different T cell subsets, activation markers, and growth factors can provide a useful measurement of the immune response to an antigen.

Central memory cells are characterized phenotypically by high expression levels of CD45R0 in humans [43,44]. Our results showed that the CD4^+^ T_CM_ subset was maintained between week 8 and week 12 as shown by co-expression of CD45R0^+^CD27^+^ on CD4^+^T cells. Conversely, non-significant declines in the CD8^+^ central memory were noted. Kalia et al., observed the maintenance of long-lived CD4^+^ central memory T cells, which are essential for long term immunological memory and protection from disease progression [45]. CD4^+^ T cells further provide help for the generation and maintenance of CD8^+^ T cell memory [30,45]. A study by Descours et al. also suggested that decreased HIV infection of T_CM_ cells in long term non-progressors was associated with HIV-specific CD8^+^ T cell responses and is a mechanism of T_CM_ preservation and lack of disease progression [46]. Memory T cells show a range of differentiated states defined by phenotype, function, anatomic localization, and contribution to protection from reinfection [47]. Memory T cell populations persist for a very long period of time, often for the lifetime of an individual, in the absence of repeated antigenic exposure [48].

In the CD4^+^ effector memory cell compartment (CD45R0^+^CD27^–^), there was a varying trend characterized by an increase at week 4, and a statistically significant decline between week 8 and week 12. The CD8^+^ T_EM_ subpopulation was observed in low numbers, albeit, with slight increases, which were not statistically significant. This is contrary to the findings of Conrad et al., who suggested that the magnitude of the HIV-specific CD8^+^ T cell response decreases with suppression of viremia [17]. However, the slight increases could be attributed to the presence of cryptococcal co-infection [49,50].

Immune activation was defined by an increase in T cells co-expressing the activation markers CD38 and HLA-DR. Consistent with a study by Serano-Villar et al., activation markers remained elevated throughout the 12 weeks [32]. Some descriptive studies report that CD8^+^ T cell activation can predict the virologic response to treatment and, hence, it is a sensitive indicator of ART activity [51,52,53].

The degree to which T cells express certain markers of cellular activation is strongly predictive of subsequent disease outcomes, independent of other factors [12,17,53]. ART reverses this effect, but this reversal is often incomplete, even after effective viral suppression [18]. The increase in HLA-DR expression associated with functional T cell defects [16] may precede the occurrence of a low CD4^+^ T cell count and shows a linear increase with increasing disease severity [2]. So far, a positive correlation between HIV-1 viremia and the expression of CD38 or HLA-DR has been described in CD8^+^ T cells only [54]. However, it was noted that chronic persistent infections are typically associated with compromised CD8 T cell function [9,55]. Strikingly, even those with recovery of near-normal CD4^+^ T cell counts may maintain chronic immune activation that has been linked to an increased risk of non-AIDS-related morbidity and mortality [11,12,56].

Levels of expression of PD-1 molecules on antigen-specific T cells in persistent infections provide a signature of functional T cell exhaustion. Upon TCR triggering, PD-1 is expressed and accumulates at the immunological synapse [9]. In this study, PD-1 was expressed at relatively high levels in CD4^+^ T cells compared to the levels in CD8^+^ T cells. It has been suggested that the accumulation of immunosenescent T cells is associated with poor CD4^+^ T cell recovery in treated HIV-infected individuals [57]. PD-1 expression on CD8^+^ T cells was associated with CD8^+^ T cell activation. Chronic HIV-1-infected patients express higher levels of PD-1 than uninfected individuals [58]. CD4^+^ T cells expressing high levels of PD-1 are enriched for HIV DNA [59], and the size of the HIV reservoir during ART correlates with the frequency of PD-1 expressing cells [13].

## 5. Conclusions

Our results suggest that 12 weeks after the initiation of ART, T cells among CM patients were partially restored with initial recovery starting four weeks after treatment, while T cell memory subsets demonstrated variable recovery over time. The gradual CD4 T cell recovery in these CM co-infected patients could decrease the incidence of immune reconstitution inflammatory events that are thought to occur if ART is initiated too early after a CM diagnosis. A gradual trend in the increase of CD4^+^ T cell subsets reveals that specific CD4^+^ T cell subsets recover while other subsets may require longer periods of ART treatment.

ART initiated during chronic HIV infection leads to an improvement in T cell responses. Our results demonstrate that ART preserves the central memory subset while suppressing viral replication and, consequently, the effector memory T cells decrease. The CD8^+^ T cell pool did not seem to be influenced by CD4^+^ T cell depletion.

## Figures and Tables

**Figure 1 jof-05-00063-f001:**
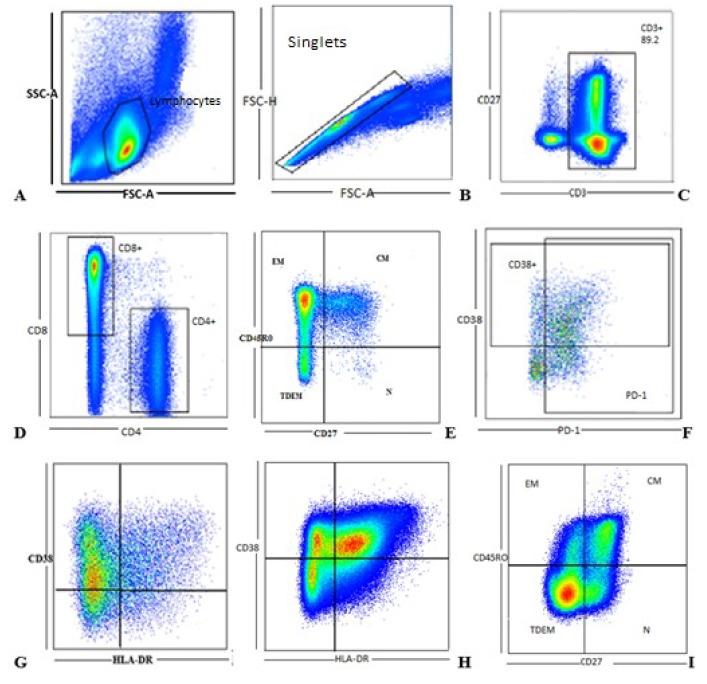
Gating strategy: Flow cytometry analysis of cryptococcal-specific CD4^+^ and CD8^+^ T cell memory and phenotypes evaluated in a 6 h assay: (**A**) We first used a singlet gate to exclude doublets by gating on forward scatter-area (FSC-A) against forward scatter-height (FSC-H). (**B**) Lymphocytes were then selected using a forward scatter-area (FSC-A) against side scatter-area (SSC-A) gate. (**C**) Conventional T cells were selected by gating on CD3^+^ cells from the total lymphocyte population, which were further divided to (**D**) CD4^+^ and CD8^+^ T cells. Memory phenotypes were assessed by variable expression of CD45R0 and CD27 on the CD4^+^ (**E**) and CD8^+^ (**I**) T cell populations; while T cell activation was assessed by expression of CD38 and HLA-DR on the CD4^+^ (**G**) and CD8^+^ (**H**) T cell subsets. T cell exhaustion was assessed by expression of PD-1 (**F**) on both populations. CM represents cryptococcal meningitis. QuaradntDEM.

**Figure 2 jof-05-00063-f002:**
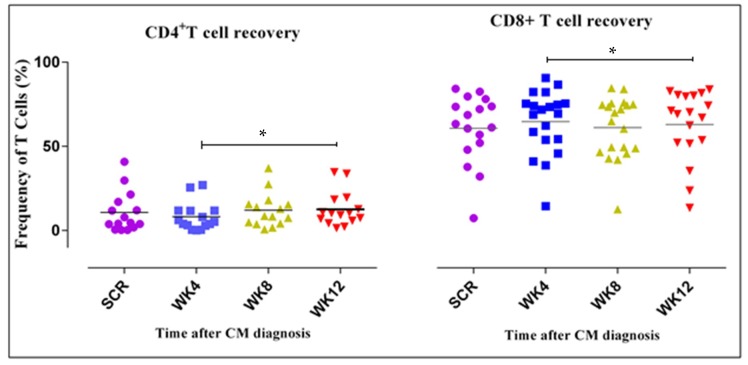
Recovery of CD4^+^ and CD8^+^ T cell populations.

**Figure 3 jof-05-00063-f003:**
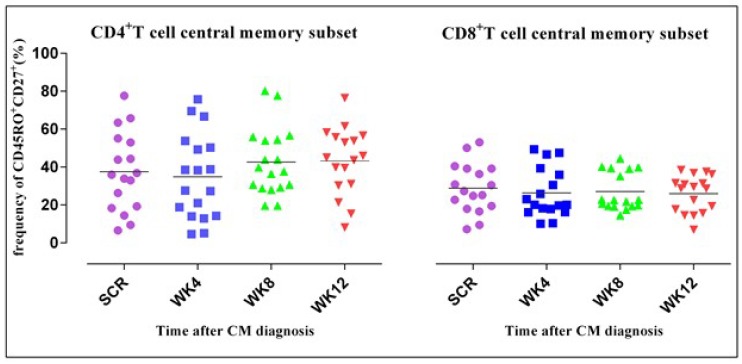
The CD4^+^ T_CM_ subset frequency remained stable yet the CD8^+^ T_CM_ subset declined.

**Figure 4 jof-05-00063-f004:**
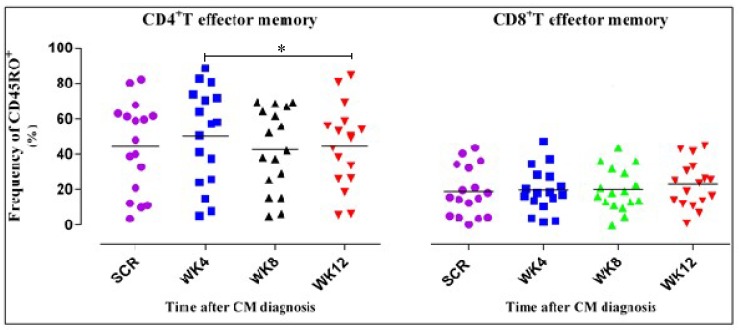
CD4^+^ T_EM_ subset expression shows significant decline contrary to the CD8^+^ T_EM_ subsets.

**Figure 5 jof-05-00063-f005:**
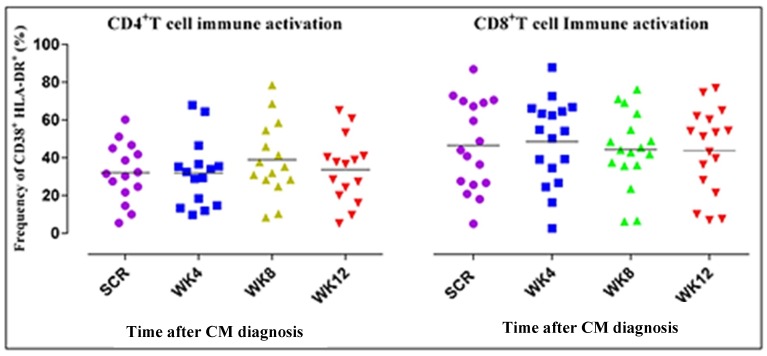
Immune activation levels remained elevated with of the T cell subsets.

**Figure 6 jof-05-00063-f006:**
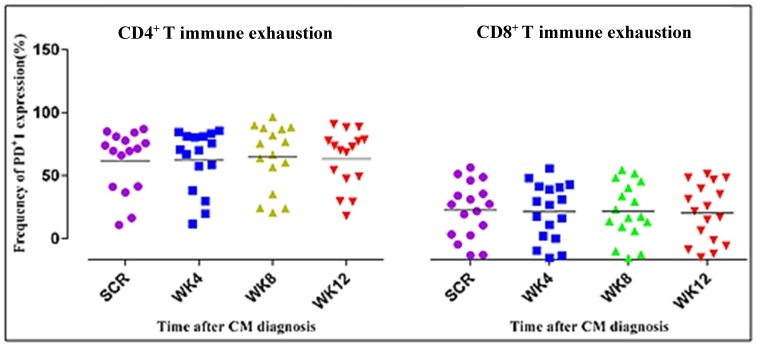
Immune exhaustion levels remained elevated for CD4^+^ and CD8^+^ T cell subpopulations.

**Table 1 jof-05-00063-t001:** Cell surface markers used as parameters to define T cell phenotypes.

Cell Marker	Phenotype Function
CD3	T cell lineage marker
CD4	CD4^+^ T lineage
CD8	CD8^+^ T lineage
CD38	Immune activation
CD45R0	T cell memory
CD27	T cell memory
PD-1	Immune exhaustion
HLA-DR	Immune activation

**Table 2 jof-05-00063-t002:** Baseline characteristics of the study participants.

**Mean Age, Years**	37
**Female, *n* (%)**	7 (35%)
**Male, *n* (%)**	13 (65%)
**Baseline CD4 T cell count, mean (SD)**	34 (±32) cells/mm^3^
**Viral load (log_10_), Mean (SD)**	6.33 ± 0.66

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
