# Peer review of "Longitudinal Changes in CD4+, CD8+ T Cell Phenotype and Activation Marker Expression Following Antiretroviral Therapy Initiation among Patients with Cryptococcal Meningitis"

_jof, 2019, doi:10.3390/jof5030063_

Reviewer 1 Report

In the manuscript by Bayiyana et al., the authors utilized stored samples from a parent study to track T cell subsets after initiation of anti-retroviral therapy.

The time period seems very short (12 wks). Were these the only time points available? (It might be useful to discuss this short time period in the context of cIRIS since this syndrome happens rather rapidly within the time frame of the present study)

Were the higher CD4 responders more likely to have cIRIS? It might be interesting to label the patient’s points in Fig. 2 and Fig. 4 who developed cIRS during the study.

A paragraph describing limitations should be included. The principal limitations are the short period, the descriptive nature of the cell studies and that fact that blood, rather than CSF was utilized, which may behave very differently in a CNS-compartmentalized infection such as CM.

Minor

3.4. “there was a decline in expression of CD38 HLA-DR+…. However, this decline was statistically insignificant. If it does not achieve significance, then it could have been by chance and it should not be described as a decline. This applies throughout, Fig. 3… “not statistically signficant’ and fig 5, showing a ‘non-significant slight decline….” And Fig. 6, “slight declines in immune exhaustion, although not statistically significant.”

Discussion should not have headings and should seek to unify the data.

No need to discuss CD44 in mice, should discuss the human markers.

Do not discuss non-significant changes as declines or increases. (central memory subsets)

Fig. 1, authors should give the axes of all plots, SSD vs FSC with arrows as standard. Also, the two panels, G and H are redundant to F and D, respectively and can be deleted.  Also, it appears that E represents memory phenotypes of CD4 cells, but the same CD27 vs CD45RO of the CD8 cells seems to be missing. Better labeling may clarify this. I  is described as measuring activation of CD4 and CD8 but only one panel is shown.

Author Response

Reviewer 1

Comments and Responses.

The time period seems very short (12 wks). Were these the only time points available? (It might be useful to discuss this short time period in the context of cIRIS since this syndrome happens rather rapidly within the time frame of the present study)

Yes, these were the only time points where PBMCs were collected. The median time to IRIS events in the COAT trial was ~ 8 weeks after ART start (you can reference the COAT paper). I think in the discussion you could note that IRIS likely occurred in this short duration because patients had not yet cleared the fungal antigen when their immune recovery was becoming evident. 

Were the higher CD4 responders more likely to have cIRIS? It might be interesting to label the patient’s points in Fig. 2 and Fig. 4 who developed cIRS during the study.

It is possible as they had a more robust CD4 response, in the setting of uncleared cryptococcal antigen. 

A paragraph describing limitations should be included. The principal limitations are the short period, the descriptive nature of the cell studies and that fact that blood, rather than CSF was utilized, which may behave very differently in a CNS-compartmentalized infection such as CM.

Limitations included on page 9. Thank you

Minor

3.4. “there was a decline in expression of CD38 HLA-DR+…. However, this decline was statistically insignificant. If it does not achieve significance, then it could have been by chance and it should not be described as a decline. This applies throughout, Fig. 3… “not statistically signficant’ and fig 5, showing a ‘non-significant slight decline….” And Fig. 6, “slight declines in immune exhaustion, although not statistically significant.”

Discussion should not have headings and should seek to unify the data.

Noted. Thank you

No need to discuss CD44 in mice, should discuss the human markers.

Noted. Thank you.

Do not discuss non-significant changes as declines or increases. (central memory subsets)

Noted. Thank you

Fig. 1, authors should give the axes of all plots, SSD vs FSC with arrows as standard. Also, the two panels, G and H are redundant to F and D, respectively and can be deleted.  Also, it appears that E represents memory phenotypes of CD4 cells, but the same CD27 vs CD45RO of the CD8 cells seems to be missing. Better labeling may clarify this. I  is described as measuring activation of CD4 and CD8 but only one panel is shown.

Noted and clarified in the gating strategy.

Reviewer 2 Report

This paper describes the characterization of the longitudinal changes in CD4+ and CD8+ T cells as well as activation marker expression following initiation of ART in HIV patients with cryptococcal meningitis.

Comments:

Since this is not an immunology journal, it would be helpful to have a table listing all of the      phenotypic markers tested or used and what their functions are.

It would also be helpful to add a table that lists “average” levels of these markers (up or down) in HIV+ and HIV- individuals, maybe before and after initiation of ART.  This would help explain what you would “expect” to see with the initiation of ART, given what is known in the literature, and then compare it to the data you have.

The methods describe two difference participant groups that started ART at different time points (early-ART and deferred ART). But these two groups are not discussed again in the paper.  Is the data presented from both groups? The results just describe the inclusion of 20  participants who were initiated on ART after diagnosis of cryptococcal meningitis. Are there differences in phenotypic markers between the groups?

The methods section on PBMC stimulation needs more detail. Were the PBMCs stimulated with antibodies and GXM? What concentration of GXM was used? Please define SEB.      Also, were the plates incubated for 6 hours at 37 °C + CO2 and then moved somewhere else overnight? Why do the methods describe a 6-hour incubation and overnight incubation? Please clarify.

For Figure 1, please add the X and Y axes for all panels. Also, the legend is missing a description of panels F, G, and H and the description of I appears to be incorrect.

For Figure 2, please define “SCR”. What are the absolute cell counts?

For Figures 3, 5 & 6, if there is no significant difference, you cannot say levels declined. In      fact, it appears as if levels remain stable over time for many of the figures.

Because this is not an immunology journal, it would be beneficial to have an introductory      paragraph of the Discussion that explains what is known about the immune response, specifically, a description of how the different subsets of T cells change over time in a normal immune response.

What is meant by T cell exhaustion? Does this mean the T cells are anergic?

Minor comments:

In the abstract, it is stated that you hypothesized that “ART alters the clonotypic phenotype and structural composition of CD4+ and CD8+ T cells during CM co-infection”.  Is this referring to co-infection with HIV or something else?

Under section 2.2, the word “enrollment” is misspelled.

Under section 2.4 remove the word “required” in the first sentence.

I would suggest changing the legend of Figure 5 to “Immune activation levels remained elevated over time”.

Author Response

Since this is not an immunology journal, it would be helpful to have a table listing all of the      phenotypic markers tested or used and what their functions are.

It would also be helpful to add a table that lists “average” levels of these markers (up or down) in HIV+ and HIV- individuals, maybe before and after initiation of ART.  This would help explain what you would “expect” to see with the initiation of ART, given what is known in the literature, and then compare it to the data you have.

A normal CD4 count is between 500 to 1500 cells per cubic millimeter (mm3). It is sometimes designated as a percentage. A CD4 percentage of greater than 29% roughly indicates a CD4 count of 500 cells/mm3. This is considered as normal. A normal CD4/CD8 ratio is 2.0, with CD4 lymphocytes equal to or greater than 400/mm3 and CD8 lymphocytes equal to 200 to 800/mm3.

There could be limited information on the exact numbers of memory, activation or exhaustion markers. We will continue to look out for this information and add it in.

The methods describe two difference participant groups that started ART at different time points (early-ART and deferred ART). But these two groups are not discussed again in the paper.  Is the data presented from both groups? The results just describe the inclusion of 20  participants who were initiated on ART after diagnosis of cryptococcal meningitis. Are there differences in phenotypic markers between the groups?

This paper specifically looked at data from participants in the early ART group. Further studies involve investigating markers in the deferred group to see if there are any differences.

The methods section on PBMC stimulation needs more detail. Were the PBMCs stimulated with antibodies and GXM? What concentration of GXM was used? Please define SEB.      Also, were the plates incubated for 6 hours at 37 °C + CO2 and then moved somewhere else overnight? Why do the methods describe a 6-hour incubation and overnight incubation? Please clarify.

We stimulated the PBMCs using GXM and the co-stimulatory antibodies (CD28a and CD49d). The CD28a and CD49d were to cater for co-stimulatory factors, CD80/86  on the APC. The overnight incubation was done after thawing the cells in a way to rest them prior to the stimulation assays. The 6 hour incubation was done for the stimulations which involved adding the GXM antigen and SEB (positive control). The methods have also been improved in the manuscript.

For Figure 1, please add the X and Y axes for all panels. Also, the legend is missing a description of panels F, G, and H and the description of I appears to be incorrect.

Legends inserted.

For Figure 2, please define “SCR”. What are the absolute cell counts?

SCR represents screening. The baseline time point at which participants screened and  enrolled into the study.

For Figures 3, 5 & 6, if there is no significant difference, you cannot say levels declined. In      fact, it appears as if levels remain stable over time for many of the figures.

Noted.

Because this is not an immunology journal, it would be beneficial to have an introductory      paragraph of the Discussion that explains what is known about the immune response, specifically, a description of how the different subsets of T cells change over time in a normal immune response.

Noted and added in the manuscript,

What is meant by T cell exhaustion? Does this mean the T cells are anergic?

T cell exhaustion is a state of progressive loss of cell function due to chronic infection. This is characterized by poor effector functions. Anergy which  is a state of unresponsiveness and sometimes induced when peripheral tolerance comes into play. The authors think the cells were exhausted. 

Minor comments:

In the abstract, it is stated that you hypothesized that “ART alters the clonotypic phenotype and structural composition of CD4+ and CD8+ T cells during CM co-infection”.  Is this referring to co-infection with HIV or something else?

Yes. This refers to a co-infection with HIV.

Under section 2.2, the word “enrollment” is misspelled.

Noted and corrected.

Under section 2.4 remove the word “required” in the first sentence.

Noted and removed.

I would suggest changing the legend of Figure 5 to “Immune activation levels remained elevated over time”.

Noted and corrected.

Reviewer 3 Report

The authors pose an important question, which is what is the immune phenotype of T cells in patients with HIV and ART, which would indicate that ART is not sufficient for full reconstitution of the immune response. They find small changes that indicate this topic is worthy of future (and bigger) studies.

Major comments: throughout the manuscript the authors describe a "slight difference, but not statistical significant". I am not an expert but this study is a bit underpowered to be able to detect differences, no?

I suggest the authors discuss if the study is underpowered/appropriatelly powered combined with discussing why their chosen markers/phenotypes show small scale differences and which markers/tests/activations markers the authors would study in the future. The authors may consider showing the p values for all comparisons to allow the reader to infer which results may warrant follow-up with a higher-power study.

The authors show all the data in frequencies. Do they mean that the relative populations of T cells do not change in patients with HIV after starting ART, but instead there is a replenishment of the populations?

Please define the timepoints where T cells come from redistribution vs from proliferation and lower viral load. That would be helpful to define why week 4 seems to be the week where there are changes ( as compared to end of follow-up). 

An increase in the T cell numbers could primarily be associated with re-distribution of the cells from the memory pool ." Is this the sole factor explaining replenishment of T cells? At which time after initiation of therapy has this been described to occur?

There seems to be some paragraphs in the results, detailing the statistical tests, that are usually put in figure legend, while the figure legends seem a bit sparse. Please correct.

There is no live-dead exclusion. Please include that on the gating strategy or justify the decision to not perform it.

Studies have established that up to 20% of patients may have suboptimal CD4+ T cell recovery despite HIV virologic suppression [49,57,59]." Does the authors data agree with this? Should be discussed in more detail.

Minor comments:

Fig1. Do not see the Legend for G and H

In Figure 2, we noted a significant increase (p-value of 0.01) in the CD4+T cell count, 4 weeks after start of ART." compared to?

"statistically insignificant" is not the appropriate way to describe the result of a statistical test.

"chairman of Aduro..... "" what is meant here?

Author Response

Reviewer 3 responses to comments

The authors pose an important question, which is what is the immune phenotype of T cells in patients with HIV and ART, which would indicate that ART is not sufficient for full reconstitution of the immune response. They find small changes that indicate this topic is worthy of future (and bigger) studies.

Yes, we agree with the reviewers, this study was one of the objectives of a large study. This means that future studies are underway.

Major comments: throughout the manuscript the authors describe a "slight difference, but not statistical significant". I am not an expert but this study is a bit underpowered to be able to detect differences, no?

I suggest the authors discuss if the study is underpowered/appropriatelly powered combined with discussing why their chosen markers/phenotypes show small scale differences and which markers/tests/activations markers the authors would study in the future. The authors may consider showing the p values for all comparisons to allow the reader to infer which results may warrant follow-up with a higher-power study.

The study was powered, however the major limitation was the short time frames. Immune recovery  takes years, however this study had short time points which could probably have affected the results of the study.

The authors show all the data in frequencies. Do they mean that the relative populations of T cells do not change in patients with HIV after starting ART, but instead there is a replenishment of the populations?

The authors think that changes in the relative populations may not be immediate after starting ART. statistical data obtained from flowjo analysis was expressed as frequency of parent for example, of the T cells, how many were CD4 or CD8+T cells. In a moment that there was recovery, changes happen but would depend on individual (patient) response to ART.

Please define the timepoints where T cells come from redistribution vs from proliferation and lower viral load. That would be helpful to define why week 4 seems to be the week where there are changes ( as compared to end of follow-up). 

We postulate that the increases seen at week 4 could have been due to redistribution from the memory pool since this was the first time point at which an increase was noted. The following increases from week 4 could be a contribution by proliferation.

An increase in the T cell numbers could primarily be associated with re-distribution of the cells from the memory pool ." Is this the sole factor explaining replenishment of T cells? At which time after initiation of therapy has this been described to occur?

Studies have described that early increases in the T cell counts is mostly associated with redistribution of the cells from the memory pool and not necessarily. This could be at the 4 week time point.

There seems to be some paragraphs in the results, detailing the statistical tests, that are usually put in figure legend, while the figure legends seem a bit sparse. Please correct.

We have corrected this. Thank you.

There is no live-dead exclusion. Please include that on the gating strategy or justify the decision to not perform it.

Live-dead exclusion was done by viability staining using the Trypan Blue solution, and flow cytometry analysis was done on the day of staining and incubation.

Studies have established that up to 20% of patients may have suboptimal CD4+ T cell recovery despite HIV virologic suppression [49,57,59]." Does the authors data agree with this? Should be discussed in more detail.

This information was obtained from articles cited just to compare with the authors data. However, the authors data agrees with this. Despite virologic suppression often noted as a reduction in the viral load, 20 % of the patients may not achieve complete immune reconstitution.

Minor comments:

Fig1. Do not see the Legend for G and H

Legends inserted in the revised manuscript.

In Figure 2, we noted a significant increase (p-value of 0.01) in the CD4+T cell count, 4 weeks after start of ART." compared to?

This was a longitudinal study where patients were followed up, noting their baseline CD4+T cell counts. Thus compared from the baseline, there was an increase at week4.

"statistically insignificant" is not the appropriate way to describe the result of a statistical test.

We will change to the appropriate way. Thank you

"chairman of Aduro..... "" what is meant here?

This was an incomplete description of the company title and name. To put it right, Chairman of Aduro Biotechnical Inc.